# Area-Scalable 10^9^-Cycle-High-Endurance FeFET of Strontium Bismuth Tantalate Using a Dummy-Gate Process

**DOI:** 10.3390/nano11010101

**Published:** 2021-01-04

**Authors:** Mitsue Takahashi, Shigeki Sakai

**Affiliations:** National Institute of Advanced Industrial Science and Technology, 1-1-1 Umezono, Tsukuba, Ibaraki 305-8568, Japan; shigeki.sakai@aist.go.jp

**Keywords:** FeFET, ferroelectric, nonvolatile, semiconductor memory, SBT

## Abstract

Strontium bismuth tantalate (SBT) ferroelectric-gate field-effect transistors (FeFETs) with channel lengths of 85 nm were fabricated by a replacement-gate process. They had metal/ferroelectric/insulator/semiconductor stacked-gate structures of Ir/SBT/HfO_2_/Si. In the fabrication process, we prepared dummy-gate transistor patterns and then replaced the dummy substances with an SBT precursor. After forming Ir gate electrodes on the SBT, the whole gate stacks were annealed for SBT crystallization. Nonvolatility was confirmed by long stable data retention measured for 10^5^ s. High erase-and-program endurance of the FeFETs was demonstrated for up to 10^9^ cycles. By the new process proposed in this work, SBT-FeFETs acquire good channel-area scalability in geometry along with lithography ability.

## 1. Introduction

Ferroelectric-gate field-effect transistors (FeFETs) comprising SrBi_2_Ta_2_O_9_ (SBT) or Ca*_x_*Sr_1-*x*_Bi_2_Ta_2_O_9_ (CSBT) ferroelectrics have unique characteristics of high endurance against at least 10^8^ cycles of program and erase operations [1,2,3,4,5,6,7,8,9,10,11,12]. CSBT is a kind of SBT family which was derived from original SBT by Sr-site substitution with Ca. The material natures of SBT [13,14,15,16,17,18,19,20,21,22,23,24,25,26,27,28,29,30,31,32] and CSBT [33,34,35,36] have been intensively studied previously. FeFETs using CSBT with about *x* = 0.2 showed larger memory windows than those with SBT [5]. The invention of long-retention FeFET was first reported in 2002 and consisted of a metal/ferroelectric/insulator/semiconductor (MFIS) stacked-gate structure of Pt/SBT/(HfO_2_)_0.75_(Al_2_O_3_)_0.25_(HAO)/Si [37]. Since then, we have investigated characteristics of (C)SBT-FeFETs [1,2,3,38,39,40,41,42,43,44], improved the device performance [4,5,6,7,8,45,46], and developed FeFET-integrated circuits [9,10,11,12,47,48,49,50,51,52]. For improving the single FeFET performance, we succeeded in reducing gate voltage (*V*_g_) from the initial 6~8 [1] to 3.3 V [8]. Another progress was in shrinking gate-metal length (*L*_m_) from the initial 10 μm [1] to 100 nm [7].

The conventional (C)SBT-FeFETs were formed by etching the gate stacks. By decreasing the FeFET gate length, SBT etching-damage problems [29,30,31,32] on the gate-stack sidewalls became significant. Since we recognized that *L*_m_ = 100 nm was approaching the shortest limit by the conventional method based on etching, we changed the fabrication strategy to shape the gate stacks from etching-down to filling-up. The new (C)SBT-FeFET process is outlined as follows: Dummy-gate transistor patterns with self-aligned source- and drain regions are prepared in advance. The dummy substance is selectively removed to leave grooves which are later filled up with SBT precursor. Gate electrodes are formed. Finally, whole gate stacks of Ir/SBT/HfO_2_/Si are annealed for SBT crystallization. In the new FeFET process, the (C)SBT sidewall of the gate stack is not exposed to etching plasma. The sidewall is thus free from etching damage problem [6]. Consequently, the ferroelectric becomes more controllable in terms of quality and more scalable in terms of geometry than by the etching. The new FeFET dimensions follow good lithography progress with an adequate height of (C)SBT to show large memory windows increasing with the ferroelectric thickness [3,43]. In this work, SBT-FeFETs with gate channel lengths *L*_ch_ = 85 nm were first reported by adopting the proposed process. Excellent characteristics were demonstrated such as 10^9^ cycle erase-program endurance and long stable retention for 10^5^ s. The endurance and retention were as good as those of the conventional (C)SBT-FeFETs formed by the gate-stack etching [1,2,3,4,5,6,7,8,9,10,11,12].

## 2. Materials and Methods

### 2.1. Device Fabrication Process

The fabrication process (schematic drawings shown in Figure 1) in this work is as follows:
Step 1: Si substrate preparation.A *p*-type Si substrate patterned with FET active areas was prepared. Local-oxidation-of-silicon (LOCOS) process was used in the patterning for device isolation. The LOCOS patterns with various channel widths (*W*) were designed in a sample chip. Areas for source-, drain- and substrate-contact holes on the Si were heavily ion-doped. Sacrificial SiO_2_ on Si was removed with buffered hydrogen fluoride.Step 2: Insulator deposition.A 5 nm thick HfO_2_ was deposited on the Si substrate by a large-area pulsed-laser deposition system (Vacuum Products Corporation, Kodaira, Tokyo, Japan) [53]. A KrF laser was irradiated on a ceramic HfO_2_ target in 15.3 Pa N_2_ ambient [54]. The substrate temperature was 220 °C.Step 3: Lithography.Electron-beam (EB) lithography was performed by spin-coating an organic resist, exposing 130 kV EB, and developing. Resist patterns 550 nm tall were left on the HfO_2_/Si. They were later used as ion-implantation mask in *Step 4* and as dummy gates in *Step 7*.Step 4: Ion implantation.HfO_2_ uncovered with resist was etched out by inductively-coupled-plasma reactive-ion etching (ICP-RIE). On the exposed Si, As^+^ ions were implanted for source and drain. The energy and dose conditions were 4 keV and 5.0 × 10^12^/cm^2^.Step 5: SiO_2_ deposition.An 830 nm thick SiO_2_ was deposited to cover the resist patterns on the substrate by 300 W rf sputtering in 0.1 Pa Ar.Step 6: Flattening SiO_2_.The SiO_2_ was etched back and flattened by ICP-RIE with 1.0 Pa Ar-CF_4_ mixed gas until tops of the resists or dummy gates were exposed.Step 7: Leaving grooves on gates.The dummy-gate substances were selectively removed by O_2_ plasma ashing. There remained grooves in a 410 nm tall SiO_2_ isolation. The grooves were located on the HfO_2_ with self-aligned source and drain regions prepared in *Step 4*. The whole chip was rapidly annealed at 800 °C in ambient N_2_.Step 8: Ferroelectric deposition.SBT precursor film was deposited to fill up the grooves by a metal-organic-chemical-vapor deposition (MOCVD) system (WACOM R&D, Nihonbashi, Tokyo, Japan). Sources of Bi(C_5_H_11_O_2_)_3_, Sr[Ta(OC_2_H_5_)_5_(OC_2_H_4_OCH_3_)]_2_ and Ta(OCH_2_CH_3_)_5_ (Tri Chemical Laboratories Inc., Uenohara, Yamanashi, Japan) were used [6]. As-deposited precursor-film thickness was estimated as 80 nm on a flat place of the substrate.Step 9: Metal deposition.Ir was deposited by rf sputtering on the SBT precursor layer. Resist mask was patterned for gate electrodes by EB lithography.Step 10: Forming gate electrodes.Ir uncovered with resist was etched out by Ar^+^ ion milling. Then, the resist mask was removed by O_2_ plasma ashing.Step 11: FeFET completed.SBT precursor was deposited again by MOCVD to cover the substrate [6]. The whole substrate was annealed for crystallization of the SBT to show ferroelectricity. The annealing condition was at 780 °C in an O_2_-N_2_ mixed gas we investigated before [8]. Finally, contact holes for gate, source, drain and substrate were formed by ultraviolet g-line lithography and Ar^+^ ion milling.

### 2.2. Reason for Using SBT in FeFET

The gate stack of MFIS should be regarded as MFI(IL)S, as shown in in Figure 2a, where F, I, IL, S are connected in series. The IL is an interfacial layer between I and S which is formed during the ferroelectric crystallization annealing process of FeFETs [8,39,55,56,57]. The main component of IL is silicon dioxide with an electric permittivity (*ε*_IL_) of *ε*_IL_ = 3.9. In the MFI(IL)S, |*P*_F_| ≈ *ε*_0_·*ε*_I_·|*E*_I_| = *ε*_0_·*ε*_IL_·|*E*_IL_| = |*Q*_S_| is satisfied in any time. The *P*_F_ is ferroelectric polarization. *E*_I_ and *E*_IL_ are electric fields in the I and the IL. The *Q*_S_ is charge area density in the semiconductor surface. The *ε*_I_ is a relative permittivity of the I. The *ε*_0_ is the vacuum dielectric constant of *ε*_0_ = 8.85 × 10^−12^ F/m. For a simplified explanation, we assumed a virtual equivalent circuit of series capacitance as drawn in Figure 2a which is expressed by |*P*_F_| ≈ |*Q*_I_| = |*Q*_IL_| = |*Q*_S_| with virtual charges *Q*_I_ and *Q*_IL_ on I and IL, respectively. In MFI(IL)S, the IL suffers from a stress of field |*E*_IL_| ≈ |*P*_F_|/(*ε*_0_·*ε*_IL_) = 8.7 MV/cm even at a small |*P*_F_| = 3 μC/cm^2^. For example, real IL thickness is 2.6 nm [8] or about 1 nm [55,56,57]. Electric-field-assisted tunnel current through such a thin SiO_2_ [58,59] brings charge injection into the gate stack from S across IL. In erase-and-program operations, a large *E*_IL_ derived from a large *P*_F_ swing induces significant trapped-charge accumulation which accelerates endurance degradations [2,52]. According to our experience [43,52,60], |*P*_F_| should normally be less than 2.5 μC/cm^2^ all the time and should not exceed 2.0 μC/cm^2^ for further high-endurance requirements of the FeFET.

Ferroelectric materials show *P*_F_ versus *E*_F_ hysteresis loops as illustrated in Figure 2b. The *E*_F_ is the electric field across the F. We defined *E*_max_ as the positive maximum *E*_F_ and *P*_max_ as the *P*_F_ at *E*_F_ = *E*_max_. Similarly, *E*_min_ and *P*_min_ are the negative minimum *E*_F_ and the *P*_F_ at *E*_F_ = *E*_min_. The loop is called “major” loop when the *E*_max_ and |*E*_min_| are strong enough to force *P*_F_ saturated, whereas it is called “minor” loop when *P*_F_ is unsaturated by moderate *E*_F_ swing. In SBT-FeFETs, restrictions of *P*_max_ ≤ 2.5 μC/cm^2^ corresponding to the minor loops are used during all operations as we emphasized in early works [39,43,52,60].

Regarding a ferroelectric hidden in MFI(IL)S, an exact symmetric swing maximum, i.e., *P*_max_ = |*P*_min_| or *E*_max_ = |*E*_min_|, is difficult because |*Q*_S_| versus *Φ*_S_ is very asymmetric [61,62]. The *Q*_S_ is the charge area density of the semiconductor surface and *Φ*_S_ is the surface potential. Presence of the flat-band voltage *V*_fb_ makes the symmetric swing further difficult. However, to simplify the physical explanation, *P*_max_ = |*P*_min_| and *E*_max_ = |*E*_min_| are assumed as shown in Figure 2b with *V*_fb_ = 0V. In every *P*_F_-*E*_F_ loop, the *E*_F_ width at *P*_F_ = 0 is defined as *E*_w_ being related with a voltage memory window (*V*_w_) by an approximate expression *E*_w_ = 2*E*_c_ = *V*_w_/*d*_F_, where the *E*_c_ is a coercive field and *d*_F_ is ferroelectric thickness. According to a method we proposed before [43], an important characteristic *E*_max_ of the ferroelectric can be evaluated which has not been measurable by direct probing on a FeFET. If *P*_max_ is provided, a gate voltage *V*_g_ to achieve a target memory window *V*_w_ = *E*_w_·*d*_F_ can be estimated as a sum of *E*_max_·*d*_F_, *E*_I_·*d*_I_, *E*_IL_·*d*_IL_ and *Φ*_S_ at *Q*_S_ = *P*_max_. An exact discussion can be found in the paper [43].

For instance, Pt/SBT/HAO/Si FeFETs showed *E*_w_ = 18 kV/cm at *P*_max_ = 2.0 μC/cm^2^ and *E*_max_ = 25 kV/cm [43]. By adopting an advanced process [8], Ir/CSBT/HfO_2_/Si FeFETs had the best improved values of *E*_w_ = 65 kV/cm at *P*_max_ = 2.0 μC/cm^2^ and *E*_max_ = 140 kV/cm [3,43]. A good reason for using (C)SBT in Si-based FeFETs is the (C)SBT ferroelectric nature of a convenient minor *P*_F_-*E*_F_ loop [14,17,20] which has *E*_w_ available and is controllable in a restricted *P*_F_ range of *P*_max_ ≤ 2 μC/cm^2^ with *E*_max_ ≤ 140 kV/cm.

There are some other ferroelectric materials also intensively studied for applications in Si-based MFIS FeFETs. Regarding Pb_5_Ge_3_O_11_ (PGO), attempts to develop replacement-gate-type Pt/PGO/ZrO_2_/Si FeFETs were reported [63] but the erase-program-test results of the FeFETs were not found although the ferroelectric itself showed a good potential *P*_max_-*E*_max_ and *E*_w_ − *E*_max_ judging from hysteresis loops of the PGO metal/ferroelectric/metal capacitors [64]. Regarding another candidate, the ferroelectric HfO_2_ family [55,56,57,65,66,67,68,69,70], the intrinsic material nature may not be suitable for applying to Si-based FeFETs. Informative minor hysteresis loops were reported on Y-doped HfO_2_ in which *E*_w_ seemed nearly equal to 0 V/cm at *P*_max_ = 2.0 μC/cm^2^, although it was as large as about 1 MV/cm at *P*_max_ = 10 μC/cm^2^ [66]. Operation of the FeFETs under the restriction of *P*_max_ ≤ 2 μC/cm^2^ may be difficult. Some reports suggested that HfO_2_-FeFETs cannot help using a large *P*_max_ (>>2 μC/cm^2^) [52,55]. The large *P*_max_ may induce significant charge injection into the gate stack. As far as we know, fair works on HfO_2_-FeFETs have not cleared 10^8^ cycles endurance in spite of using sophisticated production facilities [56,67,68,69,70].

## 3. Results and Discussion

### 3.1. Device Dimensions

A cross-sectional scanning-electron-microscope photograph of an Ir/SBT/HfO_2_/Si FeFET fabricated by the new proposed process is shown in Figure 3a. Figure 3b shows the same picture added with support lines to clarify the material boundaries. The schematic drawing of the FeFET was assigned with four terminals of gate, drain, source and substrate (Figure 3c). The gate-channel length (*L*_ch_) was *L*_ch_ = 85 nm. The gate-channel width was *W* = 100 μm depending on the initial LOCOS pattern designed in *Step 1* in Section 2.1. The metal-gate length *L*_m_ was 150 nm which could be shorter but was not the focus in this work. The SBT precursor film thickness was about 80 nm measured on a flat place. By filling gate grooves with SBT precursor (*Step 8* in Section 2.1.), the effective SBT height (*H*) was finally about 450 nm which was a distance between Ir and HfO_2_. Area scalability of the new FeFET was equivalent to that of the dummy gates which are organic resist patterns made by lithography. From the viewpoint of Si transistor technology, *L*_ch_ = 10 nm is expected to be the critical limit [71]. A significant Curie-temperature decrease in SBT started when particle were sizes of around 20 nm [25]. Thus, the prospective shortest limit of *L*_ch_ by our proposed FeFET process may be around 20 nm.

### 3.2. Electrical Characterizations

In this study, memory windows, endurance and retention of FeFETs were investigated at room temperature. A semiconductor parameter analyzer (4156C, Keysight Technologies, Santa Rosa, CA, USA) was used for measuring static drain current versus gate voltage (*I*_d_–*V*_g_) curves of the FeFETs. A pulse generator (81110A, Keysight Technologies, Santa Rosa, CA, USA) was used to apply *V*_g_ pulses. The instruments were computer-controlled using programs written by the language of LabVIEW (ver. 10, National Instruments, Austin, TX, USA).

#### 3.2.1. Memory Windows

As an elementary test of the FeFETs, *I*_d_–*V*_g_ hysteresis loops were investigated (Figure 4). The *I*_d_ was measured by *V*_g_ increments and decrements with 0.1 V steps. The *V*_g_ sweeping ranges were *V*_g_ = 1 ± 4 V, 1 ± 5 V and 1 ± 6 V. Drain voltage (*V*_d_), source voltage (*V*_s_) and substrate voltage (*V*_sub_) were fixed to *V*_d_ = 0.1 V and *V*_s_ = *V*_sub_ = 0 V during the measurements. The *I*_d_–*V*_g_ showed hysteresis loops drawn in counter-clockwise directions because the FeFET was an *n*-channel-type one. In an *I*_d_–*V*_g_ curve, threshold voltage (*V*_th_) was defined as a *V*_g_ value at *I*_d_/*W* = 1 × 10^−7^ A/cm. Two *V*_th_ values were extracted from the left- and right-side curves in an *I*_d_–*V*_g_ hysteresis loop. A memory window was defined as the *V*_th_ difference. In this work, we call this a *static* memory window (*V*_w_) because *V*_g_ sweep by 4156C is slow. The static *V*_w_ was, for instance, 1.0 V by sweeping *V*_g_ from −5 to 7 V then back to −5 V, or at *V*_g_ = 1 ± 6 V as expressed in Figure 4. During the measurement of a wide-range *I*_d_ from 10^−12^ to 10^−4^ A as indicated in Figure 4, *V*_g_ sweep speed depends on the current range. Therefore, an *I*_d_–*V*_g_ hysteresis curve only gives reference information that is not suitable for accurate discussion.

For an accurate understanding, the FeFET performance, a pulsed *V*_g_ with a controlled time width, was applied to the FeFETs for the erase (*Ers*) or program (*Prg*) operation. The *V*_g_ pulse heights with the time widths were (*V*_E_, *t*_E_) for *Ers*, and (*V*_P_, *t*_P_) for *Prg*, respectively. For the *n*-channel-type FeFET, the *V*_E_ was negative (*V*_E_ < 0 V) and *V*_P_ was positive (*V*_P_ > 0 V) [9]. The pulse time widths *t*_E_ and *t*_P_ were the same with each other in this work (*t*_E_ = *t*_P_ = *t*_EP_). After, *Er*s and *Prg*, *I*_d_–*V*_g_ curves were individually measured with a small common *V*_g_ range for *Read*. Two *V*_th_ values were defined in the *I*_d_–*V*_g_ curves as the *V*_g_ at *I*_d_/*W* = 1 × 10^−7^ A/cm. They were expressed as *V*_thE_ after *Er*s and *V*_thP_ after *Prg*. The *V*_thE_ was larger than the *V*_thP_ [9]. The *V*_th_ difference of Δ*V*_th_ = *V*_thE_ − *V*_thP_ was defined as a memory window obtained by read operation after erase-and-program pulse applications. The memory window Δ*V*_th_ is normally smaller than the above-mentioned static *V*_w_, because slow switching components in a ferroelectric do not respond to short pulses [27,72,73]. The *V*_thE_ and *V*_thP_ were investigated by repeating a series of operations: *Er*s, *Read*, *Prg*, *Read*, in this order (Figure 5a). In *Er*s, a pulsed *V*_g_ of (*V*_E_, *t*_EP_) was applied with keeping *V*_d_ = *V*_s_ = *V*_sub_ = 0 V. In *Read* after *Ers*, a *V*_thE_ was extracted from an *I*_d_-*V*_g_ curve drawn by narrow-range varying *V*_g_ from 0 to 1.1 V at *V*_d_ = 0.1 V and *V*_s_ = *V*_sub_ = 0 V. In *Prg*, a pulsed *V*_g_ of (*V*_P_, *t*_EP_) was applied, keeping *V*_d_ = *V*_s_ = *V*_sub_ = 0 V. In *Read* after *Prg*, a *V*_thP_ was extracted from an *I*_d_–*V*_g_ curve drawn under exactly the same conditions as those in *Read* after *Ers*.

Figure 5b shows *V*_thE_ and *V*_thP_ by *Read* after *Er*s and *Prg* for three sets of (*V*_E_, *t*_EP_) and (*V*_P_, *t*_EP_) of |*V*_E_| = *V*_P_ = 6, 7 and 8 V. Every marker corresponds to the measured *V*_thE_ and *V*_thP_. Memory windows, Δ*V*_th_ = *V*_thE_-*V*_thP_, as a function of pulse height |*V*_E_| = |*V*_P_| (Figure 5c) and width *t*_EP_ (Figure 5d) can be seen in Figure 5b, where the *V*_thE_ and *V*_thP_ results (not shown in Figure 5b) of other *V*_P_ (=|*V*_E_|) conditions were also used. Short *V*_g_ pulses of *t*_EP_ = 50 ns were available for *Ers* and *Prg* of the FeFET. Memory windows of Δ*V*_th_ > 0.7 V were obtained using 8 and 8.5 V pulses.

Figure 5c,d show a clear monotonic Δ*V*_th_ increases when raising either the pulse height or width. Good analog *V*_thE_ and *V*_thP_ controllability was suggested by smooth and linear Δ*V*_th_ growths with raising log(*t*_EP_) as shown in Figure 5d. The similar tendencies of Δ*V*_th_ and *t*_EP_ have already been reported in our previous works [3,5,7,9,52]. In the prior FeFETs, poly-crystalized ferroelectrics were visualized by electron backscatter diffraction (EBSD) [44]. The EBSD indicated that the (C)SBT consisted of multi-grains with various crystal orientations in the FeFETs. The poly-crystalized ferroelectrics may bring the analog *V*_thE_ and *V*_thP_ controllability to the FeFETs. In the present FeFET, there must be numerous grains in channel-width direction with *W* = 100 μm whereas a single grain or a few were expected in channel-length with *L*_ch_ = 85 nm which was smaller than average diameters of SBT grains freely grown in-plane [44].

In a preferable geometry of the replacement-gate FeFET in the future, only the channel area *L*_ch_ × *W* will be intensively scaled down with remaining the height *H*. The *H* is decided by the gate-groove depth in *Step 7* in Section 2.1 and Figure 1. The Δ*V*_th_ in this report was not yet at its best ability considering the ferroelectric height *H* = 450 nm. In the vertical direction of FeFET, a gate stack by filling SBT should be essentially the same as a large *L*_ch_ conventional one by etching SBT. Therefore, potential Δ*V*_th_ will become the same as that of conventional FeFETs by improving the details in the fabrication process in Section 2.1. An immediate target for the present FeFET will be realizing Δ*V*_th_ = 0.7 V by *Ers* of (−6V, 10 μs) and *Prg* of (6V, 10 μs) for *H* = 190 nm as demonstrated before using Pt/CSBT/HfO_2_/Si FeFETs [7].

#### 3.2.2. Retention

Retention of a FeFET was measured by the procedures as shown in Figure 6a,b. After program (*Prg*), *Retain* and *Read* were repeated during the scheduled time. In *Prg*, a *V*_g_ pulse of (*V*_P_, *t*_EP_) was applied with *V*_d_ = *V*_s_ = *V*_sub_ = 0 V. In *Retain*, all the terminals were kept at zero as *V*_g_ = *V*_d_ = *V*_s_ = *V*_sub_ = 0 V. In *Read* at a certain time *t*, an *I*_d_–*V*_g_ curve was drawn by varying *V*_g_ in a narrow range from 0 to 1.0 V at *V*_d_ = 0.1 V and *V*_s_ = *V*_sub_ = 0 V. A *V*_thP_ was extracted from the *I*_d_–*V*_g_ and plotted with a marker at *t* as shown in Figure 6c. After completing the *V*_thP_*-t*, *V*_thE_*-t* started to be measured. In erase (*Ers*), a *V*_g_ pulse of (*V*_E_, *t*_EP_) was applied with *V*_d_ = *V*_s_ = *V*_sub_ = 0 V. After *Ers*, *Retain* and *Read* were repeated during the scheduled time. The *Retain* and *Read c*onditions for *V*_thE_*-t* were the same as those for *V*_thP_*-t*. In the *Read* at a certain time *t*, an extracted *V*_thE_ was plotted with a marker at *t* as shown in Figure 6c. In this work, *V*_P_ = 8 V, *V*_E_ = −8 V and *t*_EP_ = 10 μs. The retention was measured for 10^5^ s in each of *V*_thP_*–t* and *V*_thE_*–t*. At *t* = 10^5^ s, they were still distinguishable with a difference Δ*V*_th_ = 0.26 V. When *t* > 10^3^ s, as shown in Figure 6c, the gradient of the *V*_thP_*-*log(*t*) and *V*_thE_*–*log(*t*) curves appeared to be nearly zero. A possible ten-year retention was suggested by extrapolation lines drawn on the last three markers in each branch. The present *L*_ch_ = 85 nm FeFET showed a good retention to the same extent as those of the conventional (C)SBT FeFETs [1,2,3,4,5,6,7,8,9,11,12,37,38,39,40,42,45,46,52].

#### 3.2.3. Endurance

Endurance of a FeFET was measured by the procedure shown in Figure 7a. After imposing endurance cycles on FeFETs, pairs of *V*_thE_ and *V*_thP_ were obtained. The endurance cycles consisted of periodic bipolar *V*_g_ pulses for an alternate *Ers* of (*V*_E_, *t*_EP_) and *Prg* of (*V*_P_, *t*_EP_) with *V*_d_ = *V*_s_ = *V*_sub_ = 0 V. The endurance-cycle application was interrupted at certain scheduled cycle numbers (*N*). After the *N* cycle application, *V*_thE_ and *V*_thP_ were read as follows: a series operation of *Er*s, *Read*, *Prg*, and *Read*, in this order was performed. In *Ers*, a single *V*_g_ pulse of (*V*_E_, *t*_EP_) was applied with *V*_d_ = *V*_s_ = *V*_sub_ = 0 V. In *Read* after *Ers*, an *I*_d_-*V*_g_ was measured by varying *V*_g_ in a narrow range from 0 to 1.5 V at *V*_d_ = 0.1 V and *V*_s_ = *V*_sub_ = 0 V. A *V*_thE_ was extracted from the *I*_d_*–V*_g_ and plotted with a marker at *N* as shown in Figure 7b. In *Prg*, a single *V*_g_ pulse of (*V*_P_, *t*_EP_) was applied with *V*_d_ = *V*_s_ = *V*_sub_ = 0 V. In *Read* after *Prg*, an *I*_d_*–V*_g_ was measured under the same conditions with *Read* after *Ers*. The obtained *V*_thP_ was plotted with a marker at *N* as shown in Figure 7b.

As shown in Figure 7b, the *Ers* of (−7.5 V, 10 μs) and *Prg* of (7.5 V, 10 μs) were first applied for an endurance up to *N* = 10^8^ cycles. Next, a stronger input of (−8 V, 10 μs) and (8 V, 10 μs) was applied to the same FeFET up to *N* = 10^9^ cycles. No significant sifts of *V*_thE_ and *V*_thP_ were observed throughout the measurements. By taking the minimum of the *V*_thE_ and the maximum of the *V*_thP_ in the endurance test, Δ*V*_th_ = 0.40 V for |*V*_E_| = *V*_P_ = 7.5 V and Δ*V*_th_ = 0.57 V for |*V*_E_| = *V*_P_ = 8 V were obtained. These were margins for distinguishing *V*_thE_ from *V*_thP_ as indicated in Figure 7b. In spite of using the rather complicated dummy-gate process, the *L*_ch_ = 85 nm FeFET fabricated showed high endurance up to 10^8^~10^9^ cycles. This is the same as the endurance level that (C)SBT-FeFETs inherently have [1,2,3,4,5,6,7,8,9,10,11,12].

## 4. Summary

A new fabrication process of a FeFET was proposed and demonstrated. Dummy-gate patterns with self-aligned sources and drains were prepared on a Si substrate. HfO_2_ with a thickness of 5 nm was inserted in advance between the dummy-gate substance and the Si substrate. The dummy substance was selectively removed to form a self-aligned groove on the gate. A thin SBT precursor film was deposited to fill up the groove. After forming the Ir gate electrode on the SBT, the whole gate stack was annealed for the SBT crystallization. The finished FeFET of Ir/SBT/HfO_2_/Si had a channel length *L*_ch_ = 85 nm. The FeFET exhibited a 10^9^ cycle-high endurance and long stable retentions measured for 10^5^ s. By adopting the replacement-gate process, area-scalable SBT-FeFETs with the high endurance and long retention were successfully produced.

## Figures and Tables

**Figure 1 nanomaterials-11-00101-f001:**
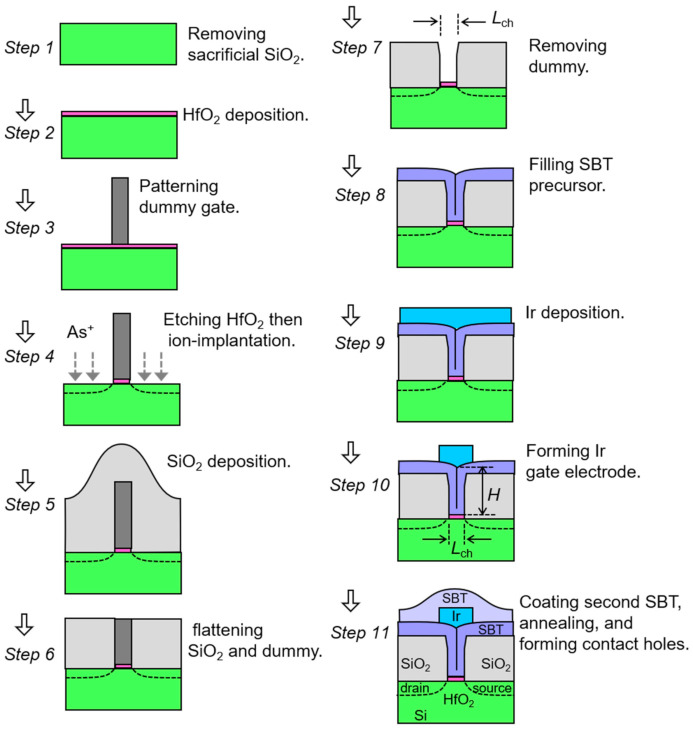
New fabrication process of Strontium bismuth tantalate (SBT)-ferroelectric-gate field-effect transistors (FeFETs) demonstrated in this work.

**Figure 2 nanomaterials-11-00101-f002:**
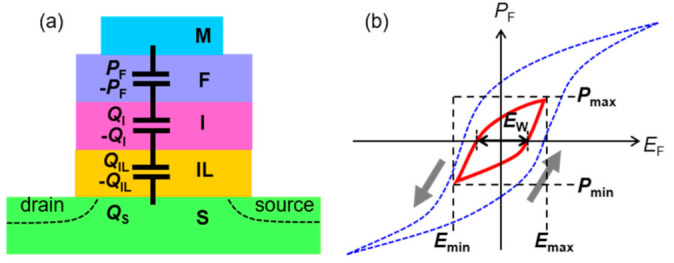
(**a**) Schematic cross-section of a FeFET with an equivalent circuit of MFI(IL)S gate stack. For convenience of explanation, the circuit is represented using virtual capacitances instead of a strict physical explanation by the electric flux density continuity, *D*. (**b**) Schematic drawings of *P*_F_ versus *E*_F_. All *P*_F_-*E*_F_ loops are drawn in counter-clockwise directions. The inner loop (red solid) is a minor loop corresponding to unsaturated *P*_F_ discussed in Section 2.2. Outer loop (blue broken) is a major loop for saturated *P*_F_ added as a reference. Every loop has its *P*_max_ at *E*_max_ and *P*_min_ at *E*_min_.

**Figure 3 nanomaterials-11-00101-f003:**
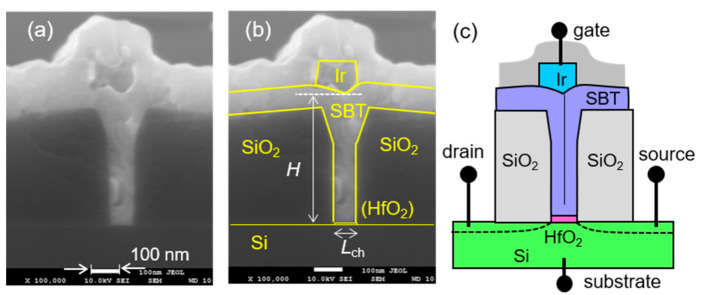
Cross-section of a FeFET with *L*_ch_ = 85 nm fabricated in this work. (**a**) Original photo by SEM observation and (**b**) the photo with supporting lines added to clarify material boundaries. (**c**) Schematic drawing assigned with gate, drain, source and substrate terminals for electrical characterizations.

**Figure 4 nanomaterials-11-00101-f004:**
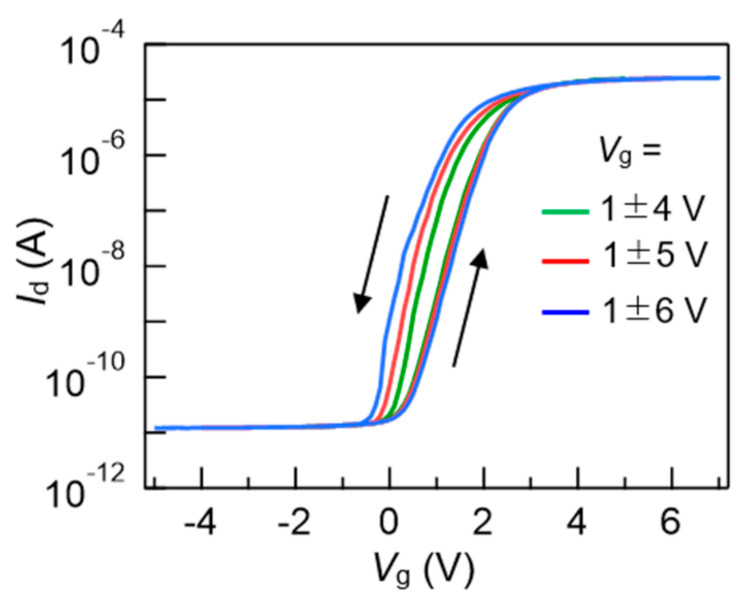
Static static drain current versus gate voltage (*I*_d_–*V*_g_) curves of a FeFET with *L*_ch_ = 85 nm. The channel width was *W* = 150 μm. *V*_g_ ranges were *V*_g_ = 1 ± 4 V, 1 ± 5 V and 1 ± 6 V.

**Figure 5 nanomaterials-11-00101-f005:**
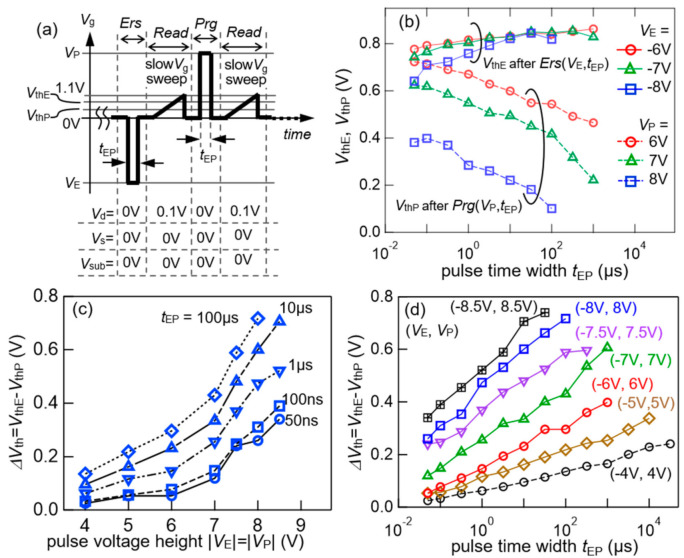
Investigation of *V*_thE_ and *V*_thP_ by applying *V*_g_ pulses to a FeFET with *L*_ch_ = 85 nm. The channel width was *W* = 100 μm. (**a**) The measurement procedure; (**b**) measured original *V*_thE_ and *V*_thP_; (**c**) pulse-height dependence of Δ*V*_th_ = *V*_thE_ − *V*_thP_ and (**d**) pulse width dependence of Δ*V*_th_.

**Figure 6 nanomaterials-11-00101-f006:**
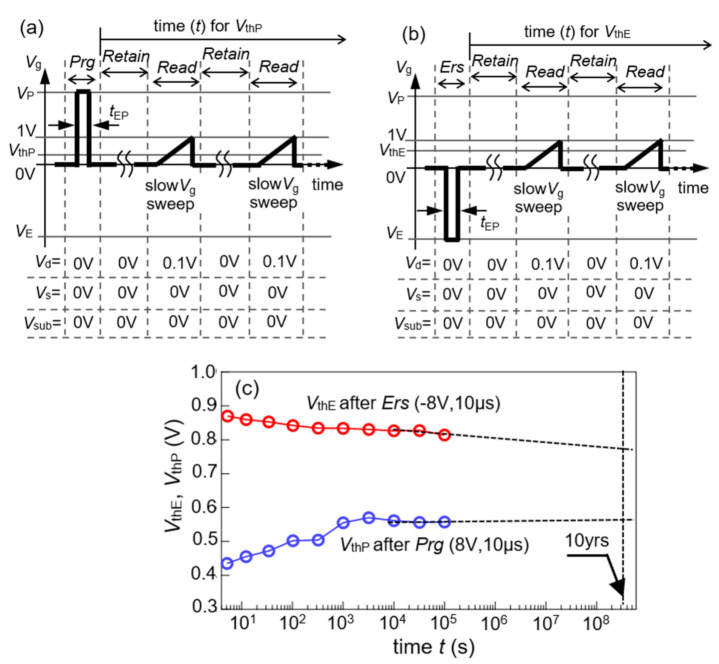
Retention investigation after applying *V*_g_ pulses to a FeFET with *L*_ch_ = 85 nm. The channel width was *W* = 100 μm. The measurement procedures for the retentions of (**a**) *V*_thP_ after *Prg* of (*V*_P_, *t*_EP_) and (**b**) *V*_thE_ after *Ers* of (*V*_E_, *t*_EP_). (**c**) The measured retentions for 10^5^ s each. Dashed lines are extrapolations of *V*_thP_*–*log(*t*) and *V*_thE_-log(*t*) for estimating *V*_thP_ and *V*_thE_ after ten years.

**Figure 7 nanomaterials-11-00101-f007:**
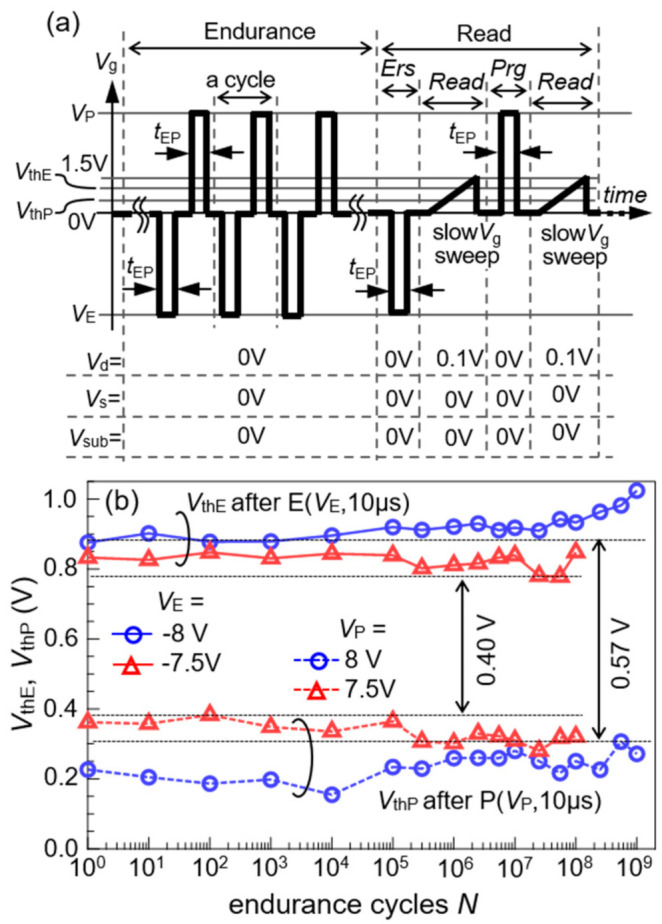
Endurance of a FeFET with *L*_ch_ = 85 nm. The channel width was *W* = 80 μm. (**a**) The measurement procedures of applying endurance cycles and reading *V*_thE_ and *V*_thP_. (**b**) Endurances were measured up to *N* = 10^8^ cycles for 7.5 V *V*_g_ pulse heights and *N* = 10^9^ cycles for 8 V.

## Data Availability

Not applicable.

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
