# Peer review of "Area-Scalable 109-Cycle-High-Endurance FeFET of Strontium Bismuth Tantalate Using a Dummy-Gate Process"

_nanomaterials, 2021, doi:10.3390/nano11010101_

Round 1

Reviewer 1 Report

In this manuscript, the authors proposed a novel method to integrate SBT into FET to make a FeFET as a memory device. The manuscript is clearly written and logically performed. It is an enjoyable experience to review this work. 

Author Response

We really appreciate your kind comments and understanding the work values.

Reviewer 2 Report

The authors reported a new process to fabricate SBT ferroelectric gate with 85 nm channel length. A dummy gate was first patterned and then replaced by SBT precursor. This method can avoid the etching damage problem on the gate sidewall and improve the device performance. I suggest the publication of this paper after addressing the following questions:

  1. What's the limit of the channel length by using the dummy-gate pattern method? Can the authors give a prediction on the minimum channel length? 
  2. If the aspect ratio of the gate size is very high, will the void be formed during the deposition of SBT gate? How can the authors resolve this problem?

Author Response

Thank you for comments and questions.

Answer 1. Regarding the minimum channel length of FeFET by the proposed method, around 20 nm may be the limit. We mentioned in the original manuscript in L.165 - L.168, as follows.

“From the viewpoint of Si transistor technology, Lch = 10 nm is expected as a critical limit [78]. Significant Curie-temperature decrease of SBT starts from about 20 nm size particle [27]. Thus, prospective shortest limit of Lch by our proposed FeFET process may be around 20 nm.”

Answer 2. As you pointed out, ordinary MOCVD techniques leave voids in the high aspect FeFETs. In order to avoid them, we actually made some improvements in SBT deposition recipe and groove-fabrication process. The details of the process may be opened in elsewhere. In this paper, we would like to report the first success of are-scalable high-endurance SBT FeFETs.

Reviewer 3 Report

In the manuscript the authors report on testing of strontium bismuth tantalate (SBT) ferroelectric gate field-effect transistors fabricated by a replacement-gate process. The resulting transistors have good retention and fatigue properties. The manuscript is well written, the manufacturing procedure is clearly described. The paper will be of interest for researchers working in the fields of ferroelectric materials and ferroelectric-gate field-effect transistors.

The disadvantage of the manuscript is a very large number of citations, especially self-citations: out of 80 citations, 38 were self-citations. Most of these citations are superfluous. I recommend that the authors greatly reduce the number of the citation leaving only significant ones, especially for Refs. 1-14 and 40-61.

Author Response

Thank you for comments. Every reference has some meaning in the manuscript. But we agreed to reduce self-citations. The seven references of 2, 6, 48, 50, 53, 54, and 56, were eliminated. The other representative works were left and renumbered. The new reference numbers in the revised manuscript are highlighted in yellow.